# COWAVE: A labelled COVID-19 wave dataset for building predictive models

**Melpakkam Pradeep**[1], **Karthik Raman**[2,3,4]*

**1** Department of Chemical Engineering, Indian Institute of Technology (IIT) Madras, Chennai, India, **2** Centre for Integrative Biology and Systems mEdicine (IBSE), IIT Madras, Chennai, India, **3** Robert Bosch Centre for Data Science and Artificial Intelligence (RBCDSAI), IIT Madras, Chennai, India, **4** Bhupat and Jyoti Mehta School of Biosciences, Department of Biotechnology, IIT Madras, Chennai, India

* kraman@iitm.ac.in

**Data Availability Statement:** The data underlying the results presented in the study are available from the GitHub repository (https://github.com/RamanLab/COWAVE).

**Funding:** Funding support from IBSE and RBCDSAI is gratefully acknowledged.

## Abstract

The ongoing COVID-19 pandemic has posed a significant global challenge to healthcare systems. Every country has seen multiple waves of this disease, placing a considerable strain on healthcare resources. Across the world, the pandemic has motivated diligent data collection, with an enormous amount of data being available in the public domain. In this manuscript, we collate COVID-19 case data from around the world (available on the World Health Organization (WHO) website), and provide various definitions for waves. Using these definitions to define labels, we create a labelled dataset, which can be used while building supervised learning classifiers. We also use a simple eXtreme Gradient Boosting (XGBoost) model to provide a minimum standard for future classifiers trained on this dataset and demonstrate the utility of our dataset for the prediction of (future) waves. This dataset will be a valuable resource for epidemiologists and others interested in the early prediction of future waves. The datasets are available from https://github.com/RamanLab/COWAVE/.

## Introduction

The COVID-19 pandemic has been a significant threat to global public health, and has continued to spread throughout the world, wreaking havoc on the healthcare infrastructure [1–5]. Despite the widespread availability of numerous vaccines and treatment methods [6–8], new outbreaks have continued to occur. With the absence of a permanent cure, predicting the onset of waves is a very important problem, since it could allow countries to prepare their health infrastructure and policies to mitigate the disastrous effects of a COVID-19 wave.

Multiple approaches have been attempted when trying to predict COVID-19 waves. The pandemic has been notoriously difficult to predict accurately [9]. Yet, the most common approach has been modelling the disease using a susceptible-infected- removed (SIR) model or its variants [10]. Various methods of computing the parameters involved in the differential equations of populations, lead to different predictions of waves [11–13]. Some models have used Bayesian Learning to estimate these parameters [14, 15]. The computation of the reproduction number ($R_0$) forms a key challenge in nearly all of these methods. However, for all these models, the evaluation typically requires labelled data. Machine Learning classifiers can also be used to predict waves, without having to estimate the various parameters of the SIR model.

**Competing interests:** The authors have declared that no competing interests exist.

In this manuscript, we provide a cleaned labelled dataset, 'COWAVE'. Our dataset contains labels on whether each day was part of a "wave" or not, along with multiple helpful features extracted from the data. We also provide a baseline supervised classifier that can be helpful when comparing the performance of any classifier trained on COWAVE. We believethat scientists would find COWAVEhelpful, for building models that learn the dynamics of various kinds of COVID-19 outbreaks, and use it to predict new waves of the pandemic, quickly and reliably. The rest of this manuscript is organised as follows: we first present the data source and methods for smoothing the data and defining waves for labelling the dataset. We then present the datasets generated in this study. We then go on to discuss example classifiers and their predictions, along with possibilities for feature generation and selection.

## Data and methods

In this section, we describe the source of our data, various smoothing algorithms and our approach to defining waves. All codes used in this study, and the datasets generated, are available from GitHub (https://github.com/RamanLab/COWAVE/). The codes were all written in Python 3, and the analyses were performed on the cloud, using Google Colaboratory (https://colab.research.google.com/).

### Source and cleaning

All data were obtained from the World Health Organization (WHO) website (https://covid19.who.int/WHO-COVID-19-global-data.csv). The columns of the dataset provide the Date Reported, the Country Name and Code, and the WHO Region, along with the new and cumulative cases and death counts. No preprocessing was performed. Two things must be noted. The first is that the new case count suddenly becomes 0 or very low for certain dates. This is likely due to no testing being done on those days/improper testing. A moving average can be used to interpolate these points, but we do not do so since smoothing will be done later.

The second thing to note is that the Country Code for Namibia is **NA**. This could cause certain libraries to interpret these strings as NaN values during the exploratory analysis stage.

### Smoothing algorithms

**Locally weighted scatter-plot smoothing.** Locally Weighted Scatter-plot Smoothing (LOWESS) is a common method used for the smoothing of scatter-plot data. It is a method equivalent to the Savitzky-Golay filter and was rediscovered by Cleveland [16]. Based on a smoothing parameter $\alpha$, the dataset is divided into subsets. A low-degree polynomial (linear, quadratic usually) is fitted locally to each subset. Points closer to the point whose response is estimated are given higher weights.

The only problem with LOWESS is that it does not provide the polynomials used to approximate the dataset. In a way, it is a black box. However, this is not a problem here since we are only interested in smoothing the data.

**Simple exponential smoothing.** The simplest of the exponential smoothing methods is called Simple Exponential Smoothing (SES). This method is suitable for forecasting data with no clear trend or seasonal pattern [17]. This also makes it an excellent method for smoothing data. Unlike a moving average where the weights are fixed, when smoothing the data by forecasting the next data point using the past data points, SES uses weights that exponentially decrease, as data points come from further in the past.

$$\hat{y}_{T+1|T} = \alpha y_T + \alpha(1-\alpha)y_{T-1} + \alpha(1-\alpha)^2 y_{T-2} + \dots \tag{1}$$

where $\hat{y}_{T+1|T}$ is the forecast for t = T+1 based on the data till t = T, $y_x$ is the data point at t = x and $\alpha$ is a smoothing parameter ($0 \leq \alpha \leq 1$).

## The wave definitions for labelling

The literature does not offer a universal definition of a "wave". While waves are characterized by rising and falling parts, which when put together form the wave, this definition is very vague for obvious reasons.

For example, in Fig 1, visually, we can find one large wave. Simply looking for rising and falling parts leads us to multiple "waves" since the smaller fluctuations (outbreaks) that may or may not be significant also are taken into account.

The notion of a wave in the context of creating a dataset for a classifier is even more ill-defined [18, 19]. Not only must the definition be reliable, but it also must only capture what can visually be observed as a "wave". To this end, we use a number of smoothing algorithms, which we have described in the previous section.

**Definition 1: Peaks and troughs (Hale).**   Hale *et al.* [20] defined waves as regions between two peaks or troughs, for a time series, after LOWESS smoothing, provided the peaks/troughs are at least one month apart. COVID waves, however, frequently have smaller "outbreaks", that aren't quite waves but get captured as waves in this definition. These characteristics can be seen in Fig 2a and 2b, S1 and S2 Figs. Here, different colours indicate different waves.

In Fig 2b, visually we find two "waves", however, this definition gives us four waves! A similar problem is found with S2 Fig, where we expect two "waves" (or three, if you count the trailing end) but find six waves. Clearly, any smaller outbreaks cause this definition to break down.

From the point of view of building a dataset for a classifier, this definition presents an even bigger problem. Since the time series can always be partitioned into sections of data between minima, there is not a single stretch of data that we can say is not part of a wave!

This can partly be avoided by imposing a minimum number of cases that a day must cross to be considered a part of a wave. For example, if we considered 200 cases as the threshold, we

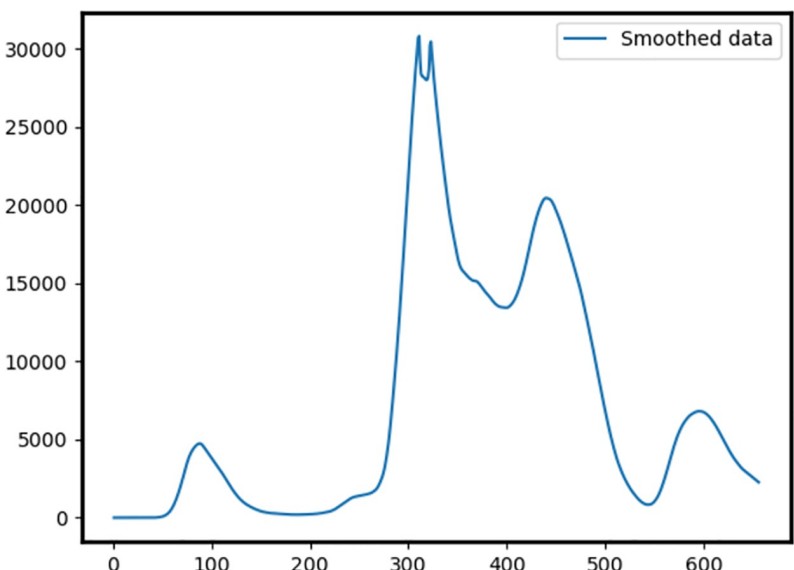

**Fig 1. LOWESS smoothed cases data for Italy from 3-Jan-2020 to 20-Oct-2021.**

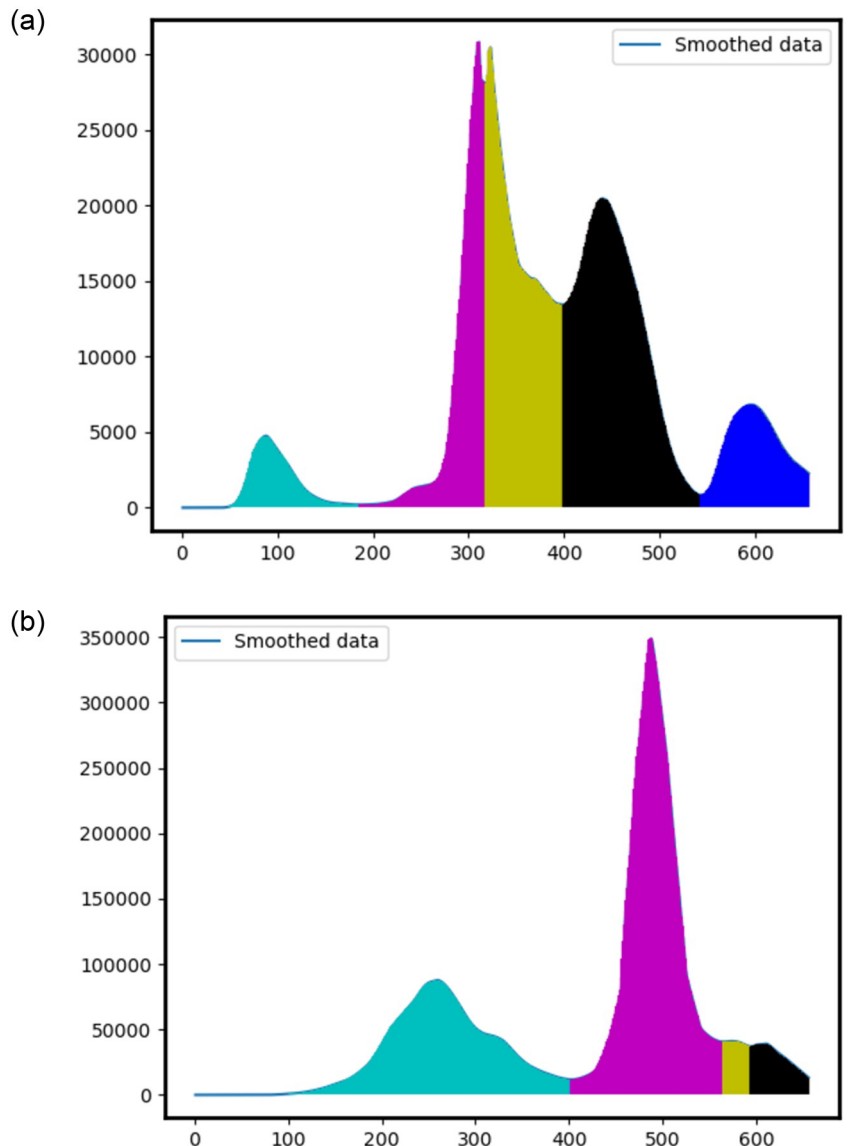

**Fig 2.** (Definition 1) LOWESS smoothed cases data for (a) Italy, (b) India. The Y-axis shows the number of cases, for a given date. The X-axis ranges from 3-Jan-2020 (Day 0) to 20-Oct-2021 (Day 655). Here, regions of different colours indicate different unique waves.

could eliminate the first cyan and purple portions in S1 Fig as a wave. However, even with a high threshold of 20 (given the maximum is 80, this is quite high a threshold), our problems would still persist, as in the case of S2 Fig, our definition presents five waves (the purple outbreak is entirely eliminated, along with some sections of cyan, yellow, black, etc.).

**Definition 2: Rapid doubling.** While the previous definition is not wrong, it does not take into account the rate at which cases rise during a certain period. Waves are mainly characterized by a rapid rise until reaching the maximum, so taking this into account could lead to much better results. Outbreaks rarely have a rapid rise. While they would be included as a "Wave" in the previous definition (due to this stretch being between two minima), in this definition, they would not. This solves the no "Non-Wave" problem to a certain degree.

Here, we find the doubling rate and normalize it:

$$\text{Doubling rate} = \frac{1}{\ln 2} \cdot \ln \left| \frac{x_t}{x_{t-7}} \right| \tag{2}$$

where $x$ is the data point (deaths, cases, hospitalizations, etc.).

We then set a threshold for the doubling rate and label it as "Wave" if the doubling rate at that time step is greater than the threshold value. To generate Fig 3a–3c we find the doubling rate. Since we use the raw data for this, there are multiple fluctuations in the doubling rate graph. So, we smooth the doubling rate using LOWESS (1/14).

All data is then normalized by dividing by the maximum of that quantity, to help fit all quantities in the same graph. The threshold is set to $\frac{d_t}{d_{t,max}} \geq 0.02$, where $d_t$ is the doubling rate at that time instant and $d_{t,max}$ is the maximum doubling rate for the whole time series. For days where the normalized doubling rate is greater than the threshold, we give the label "wave" (1), and for the rest, we give the label "non-wave" (0). From the figures that follow, we can observe some problems with this definition.

The first problem is that the number of waves labelled is not what is visually apparent. For example, in S3 Fig, we can visually see two waves, but the labelling definition gives us 15 waves! Secondly, near the maxima, the doubling rate dramatically decreases. This means our definition would never capture parts of the wave near the peak. Even with this definition, there are very few stretches of "non-wave"s. Visually, we can see that the wave and non-wave data points must not be very unbalanced. So, this is another problem. Also, outbreaks tend to have higher doubling rates than "waves" since outbreaks occur over shorter time periods. This definition would definitely label outbreaks as "waves".

**Definition 3.** We can observe that characterizing waves by doubling rate or minimas is not very helpful. What characterizes a wave, visually, is a combination of how much deviation there is from the level of the data, and over how many days it occurs. Waves are typically over several weeks, while outbreaks are over several days, regardless of the cases/deaths/hospitalizations in their respective outbreaks and waves. So by smoothing the time series, we can eliminate most outbreaks, allowing us to focus only on the waves.

In this definition, we think of waves as a deviation from the mean/level of the data. So, we subtract the mean of the complete time series of a measure for each country from each data point of the measure. Now, outbreaks are also deviations from the mean. To eliminate them, we perform two smoothing operations on this data, first LOWESS Smoothing (1/14) and then, Simple Exponential Smoothing. We can then scale this data appropriately (dividing by max of data or by standard deviation) to fit it in graphs when plotting.

After this, we label all transformed data points $<0$ as "waves" (1) and the rest as "non-waves" (0). So, the labelling definition is,

$$f(x_t) = \begin{cases} 1 & \text{if } x_t - x_{t,mean} > 0 \\ 0 & \text{else} \end{cases} \tag{3}$$

where $x_t$ is the number of cases/deaths/ hospitalizations on day t, $x_{t,mean}$ is the mean of the number of cases/deaths/ hospitalizations for that country. Since the initial parts of waves may not be captured, we apply a correction factor. We divide the labelled "wavelength" by a factor $k$ and retroactively label all points within length $\frac{\text{wavelength}}{k}$ from the start of the labelled wave as "wave" (1). The current value is $k = 6$ but different values can be experimented with for better results.

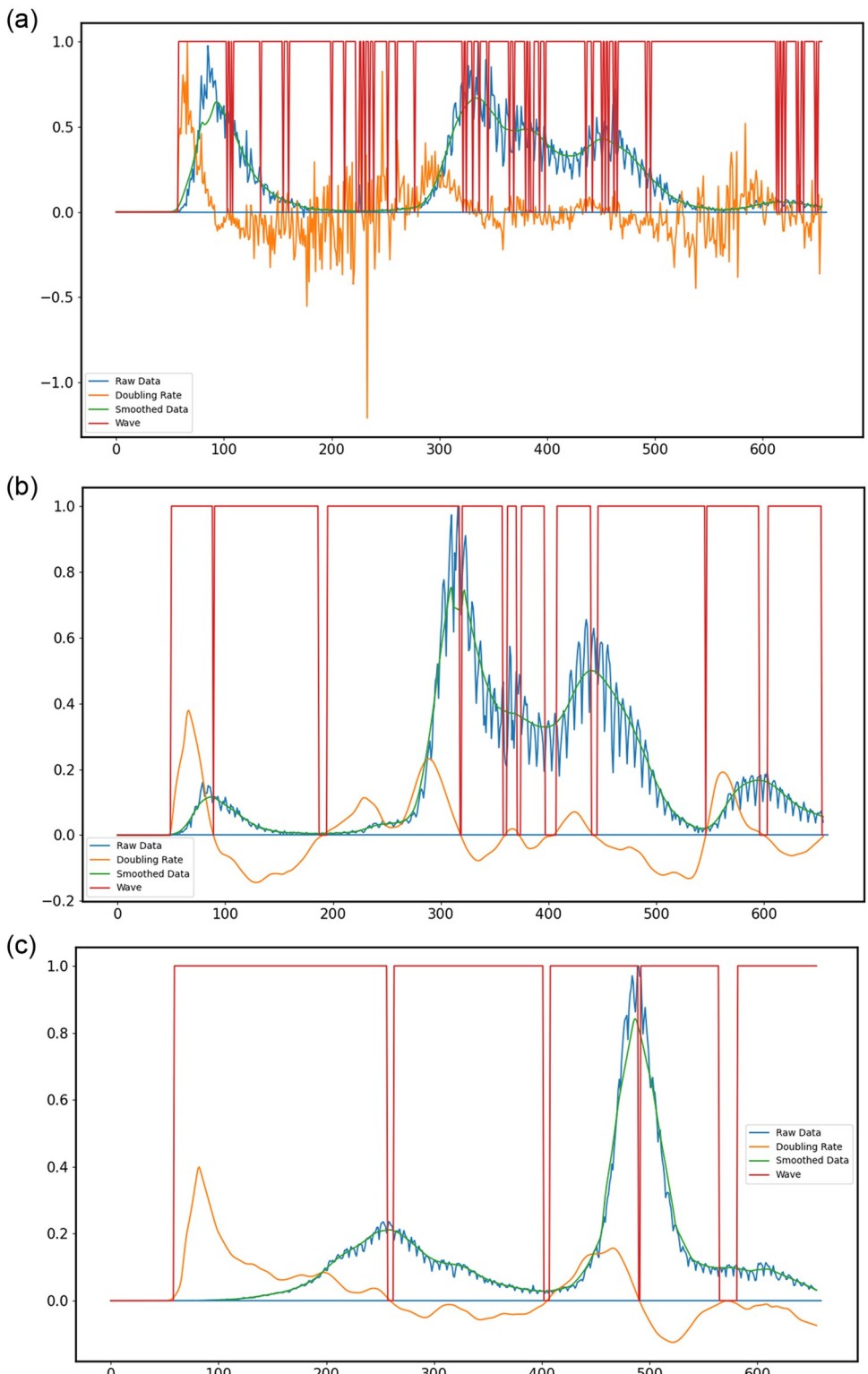

**Fig 3.** Cases data (a) without smoothing for Italy, (b) with smoothing for Italy, (c) with smoothing for India. (Definition 2) The Y-axis shows the number of cases, for a given date. The X-axis ranges from 3-Jan-2020 (Day 0) to 20-Oct-2021 (Day 655). The blue curve represents the raw data; the green curve smoothed data; and the red curve indicates whether the day is part of a wave. (a) The orange curve indicates the raw doubling rate, visually, two separate waves are observed, however, the red curve indicates more than ten waves. (b) The orange curve indicates the

smoothed doubling rate, visually, two separate waves are observed, however, the red curve indicates ten waves. (c) The orange curve indicates the smoothed doubling rate, visually, two separate waves are observed, however, the red curve indicates four waves.

In this definition, what we see is in near-perfect agreement with the labelled waves in Fig 4a and 4b, S5 and S6 Figs! One drawback is the fact that for certain waves, it appears that the wave labelling starts too late. However, for waves with initial rapid rises (most likely to overwhelm medical infrastructure), the labelling is extremely close to the start of the wave.

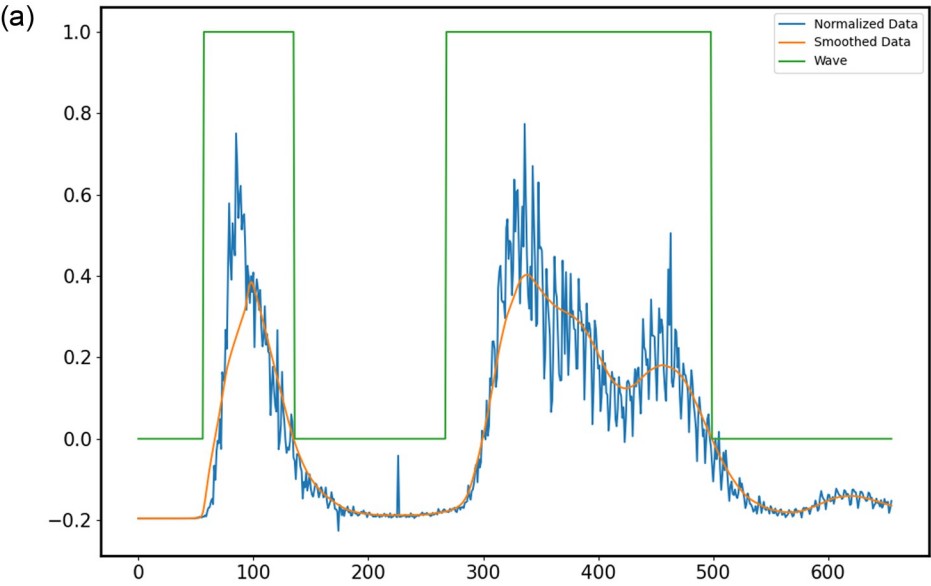

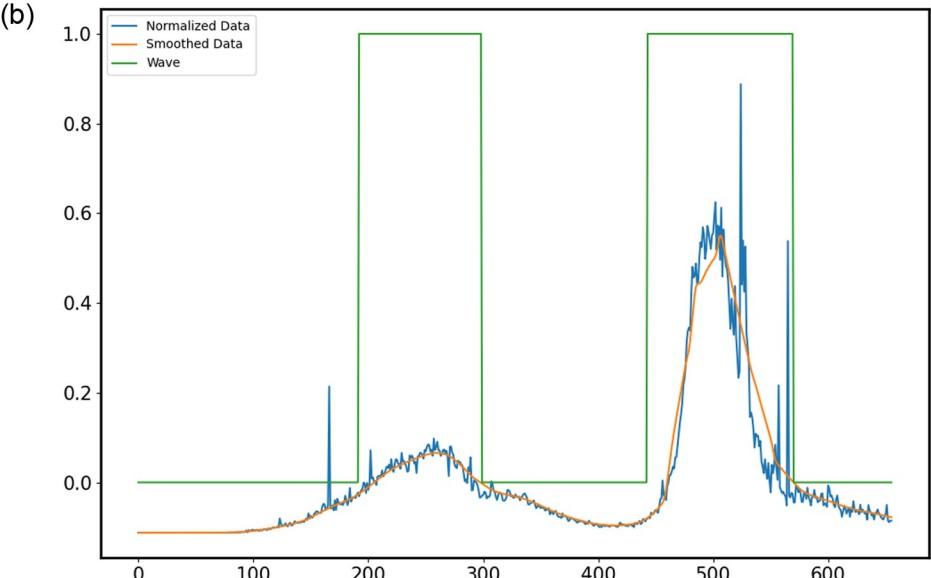

**Fig 4.** (Definition 3) LOWESS smoothed cases data for (a) Italy, (b) India. The Y-axis shows the number of cases, for a given date. The X-axis ranges from 3-Jan-2020 (Day 0) to 20-Oct-2021 (Day 655). The blue curve represents the raw data; the orange curve, smoothed data, and the green curve indicates whether the day is part of a wave. Visually, two separate waves are observed, and the green curve also indicates two waves for both (a) and (b).

We also obtained significant "non-wave" portions, as expected. This definition could be further improved by changing $k$ or subtracting $x \cdot$ mean, where $0 < x < 1$ instead of just the mean, but we shall use this definition as our labelling rule for the datasets.

## Results

We now present the datasets that we generate in our study, as well as the results of a few classification experiments, to provide baselines for further analysis and experimentation.

### The datasets

We present curated datasets in two forms, for the convenience of analysis.

**Dataset v1.** This dataset consists of four columns: Date_reported, Country_code, New_cases and Wave. The Wave column is generated as described in Definition 3, with 1 standing for the day being part of a wave, and 0 meaning the day is not part of a wave.

**Dataset v2.** This dataset consists of stretches of waves and non-waves. Each row contains the start date of the wave/non-wave (Date), the country code (Country_code), whether it is a wave or non-wave (Wave), and the case data for that stretch of wave/non-wave (as a list) (Cases).

**Training and test datasets.** For the classifiers built in the next section, the training dataset consists of the WHO cases data for all countries between country codes **AF** (*Afghanistan*) and **QA** (*Qatar*) from 3/1/2020 to 27/5/2022. The test dataset consists of the WHO cases data for all countries between country codes **RK** (*Republic of Korea*) and **ZW** (*Zimbabwe*) from 3/1/2020 to 27/5/2022. All metrics are reported for the full test and training datasets. They may vary for different subsets of these datasets. We assume that the distributions of cases do not depend on the country code assigned by WHO.

### A naïve baseline classifier

The baseline classifier is a simple classifier that we can use to compare all other classifiers. For this, we choose an SVM classifier [21] with a Radial Basis Function (*RBF*) kernel.

We first see how fast the untuned model's metrics vary as more data is used for every iteration. Instead of sending each day's data, we send a data vector with the current day's cases, along with the previous six days as well. In the next run, we send in a 14-day vector instead of the 7-day vector. In the subsequent run, we send in a 21-day vector. The results of the three runs are tabulated in Table 1.

**Table 1. Performance of different data vectors.** Shown are the accuracy, recall and precision for the untuned baseline classifier (SVM, *RBF* kernel) with the three data vectors described in this section.

| Dataset | *n*-day vector | Accuracy | Recall | Precision |
|---------|----------------|----------|--------|-----------|
| Train   | 7-day vector   | 0.686    | 0.154  | 0.722     |
|         | 14-day vector  | 0.698    | 0.169  | 0.807     |
|         | 21-day vector  | 0.705    | 0.184  | 0.839     |
| Test    | 7-day vector   | 0.647    | 0.146  | 0.614     |
|         | 14-day vector  | 0.651    | 0.145  | 0.641     |
|         | 21-day vector  | 0.654    | 0.156  | 0.649     |

## XGBoost provides much better classification performance

For an improved classifier, we look for an algorithm that runs quite fast and can classify tabulated data with good accuracy ($>60\%$) and high recall. It also should not require a very large amount of data. While these requirements have not been perfectly quantified here, tree ensemble methods seem to fit all these requirements. For this, we use the XGBoost algorithm [27].

We first see how fast the untuned model's metrics vary as more data is used for every iteration. Instead of sending each day's data, we send a data vector with the current day's cases, along with the previous six days as well. In the next run, we send in a 14-day vector instead of the 7-day vector. In the subsequent run, we send in a 21-day vector. The results of the three runs are tabulated in Table 2. It is evident that the XGBoost classifier gives significantly better results than the Baseline Classifier.

## Feature generation and selection

**Feature generation.** We can clearly see that while the metrics show an improvement, as a whole, as the length of the data vector is increased, the improvements are very marginal. It is likely that we will need to generate more features, not simply use the previous days' values as additional features if improvements are to be made to the performance. Some obvious features are the differences in the lags. We include the differences over the past seven days. Improvements have been observed to be marginal beyond this. Taking the maximum, minimum, mean and variance of the 21-day vector are also features that can be readily included.

We then split the time series into its trend, seasonal and residual components and use these as three additional features. Visualizing the time series in different ways by transforming it could also lead to better model performance in our classification task. We use the square, square root, logarithm and Box-Cox transformations here. Other transformations can also be used. These transformations need not be limited to the current-day data but can also be used for the lags and the differences.

We also include non-linear transformations of the current day's data as features, particularly using

$$f(t_{21}) = \frac{1}{1 + e^{-\frac{t_{21}}{\sigma(t_i)}}}$$

(4)

where $t_{21}$ is the current day's data and $\sigma(t_i)$ is the standard deviation of the 21-day vector. If the standard deviation is 0, we take the exponential term as 0.

The second non-linear transformation used is

$$f(t_{21}) = \frac{1}{\sqrt{2\pi}\sigma} e^{-\frac{(t_{21}-\mu)^2}{2\sigma^2}}$$

(5)

**Table 2. Performance of different data vectors.** Shown are the accuracy, recall and precision for the untuned improved classifier (XGBoost) with the three data vectors described in this section.

| Dataset | $n$-day vector | Accuracy | Recall | Precision |
|---------|----------------|----------|--------|-----------|
| Train | 7-day vector | 0.735 | 0.538 | 0.665 |
| | 14-day vector | 0.735 | 0.539 | 0.666 |
| | 21-day vector | 0.737 | 0.542 | 0.669 |
| Test | 7-day vector | 0.725 | 0.444 | 0.673 |
| | 14-day vector | 0.726 | 0.445 | 0.676 |
| | 21-day vector | 0.727 | 0.444 | 0.680 |

where $t_{21}$ is the current day's data, $\mu$ is the mean of the 21-day vector and $\sigma$ is the standard deviation of the 21-day vector.

We also find the range and covariance of each 21-day vector since these quantities give us a measure of the spread of the cases in the 21 days. The median and entropy for the 21-day vectors are also found, where the entropy ($E$) is given by,

$$E = -\sum_{i=1}^{21} x_i ln(x_i), \quad x_i = \frac{t_i}{\max(t_1, t_2, \ldots, t_{21})} \tag{6}$$

**Feature selection.** The generated features were used to evaluate the importance of the generated features using an XGBoost Classifier. Based on this, the top 13 features were selected. The feature names are listed here: T21, D7, MIN, MAX, Range, Sq, Median, Mean, Variance, PDF, Trend, Seasonal and Residual. This is done to reduce the training time for the models. From now, this set of features is referenced as "**Top 13**". Other feature selection methods may also be used [22–26].

## New features enable improved classification

**Untuned XGBoost classifier.** With these features, we train an XGBoost model on the dataset. With the default hyperparameters (as in the *xgboost* library), we achieve scores as shown in Table 3.

For tuning, we used Bayesian Optimization as implemented in the bayesian-optimization library [28]. Its speed, along with its ability to search the hyperparameter space, rather than discrete values only, were the reasons this method was chosen over GridSearch. We used three-fold cross-validation. The list of tuned hyperparameters, along with the tuning ranges, is given in Table 4.

Multiple runs were tested, and the best set of hyperparameters is shown here. However, due to the random nature of Bayesian Optimization, there is a chance of a better set of hyperparameters existing in a different tuning space or even in the same tuning space.

We present the hyperparameters and results for two classifiers, the first for the maximum accuracy obtained, and the second, biased towards a high recall, with acceptable accuracy (For example, models with 0.999 recall but with <0.5 accuracy would be rejected).

**High accuracy classifier.** Initially, the hyperparameter search was performed for the selected features. However, a higher accuracy was obtained for the tuned model when including all generated features. We present the chosen hyperparameters in Table 5, along with the results in Table 6, for both sets of features here. The classifier with the selected features runs twice as fast when compared to the classifier with all generated features.

**High recall classifier.** When predicting waves, the cost of missing waves is much more than wrongly predicting waves. Therefore, we tune our models for higher recall. Since strictly maximum recall is not our aim, we use the selected features (since it is faster). Based on

**Table 3. Performance of different feature sets.** Shown are the accuracy, recall and precision for the untuned XGBoost classifier [27] with two subsets of the generated features—all features as well as "Top 13" features described earlier.

| Dataset | Features Used | Accuracy | Recall | Precision |
|---|---|---|---|---|
| Train | All | 0.859 | 0.798 | 0.810 |
| | "Top 13" | 0.855 | 0.796 | 0.801 |
| Test | All | 0.830 | 0.690 | 0.804 |
| | "Top 13" | 0.831 | 0.702 | 0.800 |

**Table 4. Search space for hyperparameter tuning.**

| Hyperparameter | Value Range |
|---|---|
| learning_rate | (0.0005, 1) |
| max_depth | (1, 10) |
| min_child_weight | (1, 10) |
| gamma | (0, 3) |
| colsample_bytree | (0.001, 1) |
| num_boost_round | (100, 500) |
| reg_lambda | (0.01, 10) |
| scale_pos_weight | (1, 10) |
| subsample | (0.001, 1) |

**Table 5. Selected hyperparameters for the high accuracy classifier.** The first column contains the best hyperparameters for the "Top 13" features, as described in the previous section. The second column contains the best hyperparameters for all features generated.

| Hyperparameter | Selected Values | |
|---|---|---|
| | **"Top 13"** | **All features** |
| learning_rate | 0.25 | 0.709 |
| max_depth | 4 | 3 |
| min_child_weight | 9.5 | 2.791 |
| gamma | 2.1 | 2.106 |
| colsample_bytree | 0.9 | 0.619 |
| num_boost_round | 450 | 336 |
| reg_lambda | 5.36 | 4.515 |
| scale_pos_weight | 1.0 | 1.437 |
| subsample | 1.0 | 0.504 |

Bayesian Optimization, the hyperparameters were chosen as given in Table 7. With this tuned model, the results for the full test and train sets are given in Table 8. With a faster runtime, and better accuracy and recall, the XGBoost model is much better when compared to the tuned baseline.

## Conclusion

In this study, we present a new definition for demarcating "waves" in COVID-19 outbreaks. Our definition is simple and also agrees with typical visual interpretations of COVID-19 waves. We used this definition to label the daily case data of COVID-19 as "Wave" and "Non-Wave", to allow the use of supervised learning classifiers to predict waves. To illustrate

**Table 6. Performance of different feature sets.** Shown are the accuracy, recall and precision for the tuned XGBoost classifier with two subsets of the generated features—all features as well as "Top 13" Features described previously. The hyperparameters are as described in Table 5.

| Dataset | Feature Set used | Accuracy | Recall | Precision |
|---|---|---|---|---|
| Train | "Top 13" | 0.867 | 0.820 | 0.814 |
| | All | 0.885 | 0.878 | 0.817 |
| Test | "Top 13" | 0.834 | 0.716 | 0.799 |
| | All | 0.850 | 0.769 | 0.802 |

**Table 7. Selected hyperparameters for the high recall classifier.** The hyperparameters are selected based on the performance of the "Top 13" features described earlier.

| Hyperparameter | Value |
|---|---|
| learning_rate | 0.025 |
| max_depth | 9 |
| min_child_weight | 6.441 |
| gamma | 2.444 |
| colsample_bytree | 0.873 |
| num_boost_round | 307 |
| reg_lambda | 5.897 |
| scale_pos_weight | 8.198 |
| subsample | 0.174 |

**Table 8. Performance of different feature sets.** Shown are the accuracy, recall and precision for the tuned XGBoost classifier with the "Top 13" Features described earlier. The hyperparameters are as described in Table 7.

| Dataset | Accuracy | Recall | Precision |
|---|---|---|---|
| Train | 0.777 | 0.977 | 0.622 |
| Test | 0.794 | 0.933 | 0.646 |

possibilities, we build a naïve SVM-based classifier and an improved classifier based on XGBoost. We then go on to generate new features and select more informative features, and build classifiers that are specifically tuned for higher accuracy, or more importantly, higher recall. Overall, we believe that this dataset will motivate more people to work on this exciting problem of COVID-19 wave detection, and enable the creation of more high-performing classifiers to accurately, rapidly and reliably predict future COVID waves.

## Supporting information

**S1 Fig. LOWESS smoothed cases data for the Netherlands.** The Y-axis shows the number of cases, for a given date. The X-axis ranges from 3-Jan-2020 (Day 0) to 20-Oct-2021 (Day 655). Here, regions of different colours indicate different unique waves.
(TIF)

**S2 Fig. LOWESS smoothed cases data for Egypt.** The Y-axis shows the number of cases, for a given date. The X-axis ranges from 3-Jan-2020 (Day 0) to 20-Oct-2021 (Day 655). Here, regions of different colours indicate different unique waves.
(TIF)

**S3 Fig. LOWESS smoothed cases data for the Netherlands.** The Y-axis shows the number of cases, for a given date. The X-axis ranges from 3-Jan-2020 (Day 0) to 20-Oct-2021 (Day 655). The blue curve represents the raw data; the green curve, smoothed data; the orange curve, the LOWESS smoothed doubling rate, and the red curve indicates whether the day is part of a wave.
(TIF)

**S4 Fig. LOWESS smoothed cases data for Egypt.** The Y-axis shows the number of cases, for a given date. The X-axis ranges from 3-Jan-2020 (Day 0) to 20-Oct-2021 (Day 655). The blue curve represents the raw data; the green curve, smoothed data; the orange curve, the LOWESS

smoothed doubling rate, and the red curve indicates whether the day is part of a wave.
(TIF)

**S5 Fig. LOWESS smoothed cases data for the Netherlands.** The Y-axis shows the number of cases, for a given date. The X-axis ranges from 3-Jan-2020 (Day 0) to 20-Oct-2021 (Day 655). The blue curve represents the raw data; the orange curve, smoothed data, and the green curve indicates whether the day is part of a wave.
(TIF)

**S6 Fig. LOWESS smoothed cases data for Egypt.** The Y-axis shows the number of cases, for a given date. The X-axis ranges from 3-Jan-2020 (Day 0) to 20-Oct-2021 (Day 655). The blue curve represents the raw data; the orange curve, smoothed data, and the green curve indicates whether the day is part of a wave.
(TIF)

## Acknowledgments

We thank Dr. G. Ramadurai and other members of the IIT Madras RBCDSAI "COVID War Room Team" for several useful discussions.

## Author Contributions

**Conceptualization:** Karthik Raman.

**Formal analysis:** Melpakkam Pradeep.

**Funding acquisition:** Karthik Raman.

**Investigation:** Melpakkam Pradeep.

**Project administration:** Karthik Raman.

**Writing – original draft:** Melpakkam Pradeep.

**Writing – review & editing:** Karthik Raman.

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
