## [Decision Letter · Decision Letter 0]

12 Dec 2022

PONE-D-22-27577COWAVE: A Labelled COVID-19 Wave Dataset for BuildingPredictive ModelsPLOS ONE

Dear Dr. Raman,

Thank you for submitting your manuscript to PLOS ONE. After careful consideration, we feel that it has merit but does not fully meet PLOS ONE’s publication criteria as it currently stands. Therefore, we invite you to submit a revised version of the manuscript that addresses the points raised during the review process.

We look forward to receiving your revised manuscript.

Kind regards,

Hilal Tayara

Academic Editor

PLOS ONE

Journal Requirements:

3. Please ensure that you refer to Figures 8 and 9 in your text as, if accepted, production will need this reference to link the reader to the figure.

4. We note you have included a table to which you do not refer in the text of your manuscript. Please ensure that you refer to Tables 1, 6 and 8 in your text; if accepted, production will need this reference to link the reader to the Table.

Reviewers' comments:

Reviewer's Responses to Questions

**Comments to the Author**

1. Is the manuscript technically sound, and do the data support the conclusions?

Reviewer #1: Yes

Reviewer #2: Partly

2. Has the statistical analysis been performed appropriately and rigorously? 

Reviewer #1: Yes

Reviewer #2: N/A

3. Have the authors made all data underlying the findings in their manuscript fully available?

Reviewer #1: Yes

Reviewer #2: No

4. Is the manuscript presented in an intelligible fashion and written in standard English?

Reviewer #1: No

Reviewer #2: Yes

5. Review Comments to the Author

Reviewer #1: The manuscript is interesting; however, the following comment should be addressed:

1: All the sections and subsections must be included in the text, Such as :( Data and Methods, Smoothing Algorithms, etc.).

2: The abstract needs to be improved to do justice to the main contributions of the paper, also it contains some abbreviations that need to be explained.

3: The contribution is not stated also add it at the end of the introduction section.

4: Please add future work to the conclusion section and discuss it briefly.

5: This study suffers from a fresh literature review. It is recommended to boost the literature review of this study.

6: Please add the system specifications used for the evaluation as well as the programming language.

7: There are some typos and grammatical errors that should be corrected.

Reviewer #2: Summary:

In this work, COVID-19 case data have been collected from around the world. The regions of waves are labelled. Also, XGBoost model is used to provide a minimum standard for future classifiers trained on this dataset. In addition, the utility of the dataset for the prediction of (future) waves.

The manuscript is interesting; however, the following comment should be addressed :

Abstract :

- - - - - - - - - - -

1 – Some facts about the collected dataset need to be included in the abstract.

Introduction Section :

- - - - - - - - - - - - - - - - - - - - - -

2 – The introduction need to be extended because it is very short.

3 – The contribution should be included as a list at the end of the introduction section .

Data and Methods Section :

- - - - - - - - - - - - - - - - - - - - - - - -

4 – This section is fine. No comments.

The Datasets Section :

- - - - - - - - - - - - - - - - - - - - - -

5 – More details about the dataset should be included .

6 – Visual analysis should be discussed thoroughly .

Results Section :

- - - - - - - - - - - - - - - - - - - - - -

7 – The author mentioned that supervised classifier can be used. Thus , the author need to refer to other types of feature extraction tools. For example: a) 10.1109/ACCESS.2022.3170893, b) 10.1002/cpe.7311, and c) 10.3390/sym14040715 .

Conclusion Section :

- - - - - - - - - - - - - - - - - - - - - -

8 – This section is fine. No comments .

General Comments:

- - - - - - - - - - - - - - - - -

9 - There are some grammatical errors that should be corrected . It is highly recommended to be proofed the manuscript carefully .

- - - - - - - - - - - - - - - - - - - - - - - - - - - - - - - - - - - - - - - - - - - - - - - - - - - - - - - - - - - - - - - - - - - - - - - - - - - - - - - - - - - - - - - - - - - - - - - - - - - - - - - - - - - - - - - - - - - - - - - - - - - - - - - - - - - - - - - - - - - - - - - - - - - - - - - - - - - - - - - - - - - - - - - - - - - - - - - -

6. PLOS authors have the option to publish the peer review history of their article (what does this mean?). If published, this will include your full peer review and any attached files.

Reviewer #1: No

Reviewer #2: No

---

## [Author Response · Author response to Decision Letter 0]

10 Jan 2023

Response to Reviewer 1

1. All the sections and subsections must be included in the text, Such as :( Data and Methods, Smoothing Algorithms, etc.).

We have added all sections and subsections. Further, we have now (slightly) reorganized the sections and subsections for better readability.

2. The abstract needs to be improved to do justice to the main contributions of the paper, also it contains some abbreviations that need to be explained.

We have improved the abstract, with more details regarding the contributions made, the nature of the dataset created, and the baseline classifiers. All abbreviations used in the abstract, have also been explained.

3. The contribution is not stated also add it at the end of the introduction section.

We have added the contributions in the Introduction section (lines xx-yy).

4. This study suffers from a fresh literature review. It is recommended to boost the literature review of this study.

We have now included an additional paragraph in the “Introduction” section to boost the literature review, as per the reviewer’s suggestion (lines 8-13).

5. Please add the system specifications used for the evaluation as well as the programming language.

Since we are not reporting any runtimes, we have not reported system specifications. We have now clearly indicated the programming language, and that the codes were run on Google Colaboratory, at the beginning of the Data and Methods section (lines 28-32).

6. This study suffers from a fresh literature review. It is recommended to boost the literature review of this study.

We have now extended the Introduction section, with some methods of predicting COVID waves (lines 8-16).

7. There are some typos and grammatical errors that should be corrected.

We have carefully proofread the document again, and corrected all typos and grammatical errors that were found.

Response to Reviewer 2:

1. Some facts about the collected dataset need to be included in the abstract.

We have added the source of the initially collected dataset, in the abstract.

2. The introduction needs to be extended because it is very short.

We have now extended the Introduction section, with some methods of predicting COVID waves.

3. The contribution should be included as a list at the end of the introduction section.

We have included the contribution at the end of the introduction (lines 18-21).

4. This section is fine. No comments.

We thank the reviewer for their appreciation.

5. More details about the dataset should be included.

We have now included more details about the dataset used to build the COWAVE datasets.

6. Visual analysis should be discussed thoroughly.

We have improved the discussion of the visual analysis.

7. The author mentioned that a supervised classifier can be used. Thus, the author needs to refer to other types of feature extraction tools. For example: a) 10.1109/ACCESS.2022.3170893, b) 10.1002/cpe.7311, and c) 10.3390/sym14040715.

We have now cited other possible feature selection techniques (lines 223-224).

8. This section is fine. No comments.

We thank the reviewer for their appreciation.

9. There are some grammatical errors that should be corrected. It is highly recommended to be proofed the manuscript carefully.

We have carefully proofread the document again and corrected all typos and grammatical errors that were found.

---

## [Decision Letter · Decision Letter 1]

24 Mar 2023

COWAVE: A Labelled COVID-19 Wave Dataset for Building

Predictive Models

PONE-D-22-27577R1

Dear Dr. Raman,

We’re pleased to inform you that your manuscript has been judged scientifically suitable for publication and will be formally accepted for publication once it meets all outstanding technical requirements.

Kind regards,

Hilal Tayara

Academic Editor

PLOS ONE

Additional Editor Comments (optional):

Please address all comments raised by reviewr 4.

Reviewers' comments:

Reviewer's Responses to Questions

**Comments to the Author**

1. If the authors have adequately addressed your comments raised in a previous round of review and you feel that this manuscript is now acceptable for publication, you may indicate that here to bypass the “Comments to the Author” section, enter your conflict of interest statement in the “Confidential to Editor” section, and submit your "Accept" recommendation.

Reviewer #2: All comments have been addressed

Reviewer #3: All comments have been addressed

Reviewer #4: (No Response)

Reviewer #5: All comments have been addressed

Reviewer #6: All comments have been addressed

2. Is the manuscript technically sound, and do the data support the conclusions?

Reviewer #2: Yes

Reviewer #3: Yes

Reviewer #4: (No Response)

Reviewer #5: Yes

Reviewer #6: Yes

3. Has the statistical analysis been performed appropriately and rigorously? 

Reviewer #2: Yes

Reviewer #3: Yes

Reviewer #4: (No Response)

Reviewer #5: Yes

Reviewer #6: Yes

4. Have the authors made all data underlying the findings in their manuscript fully available?

Reviewer #2: Yes

Reviewer #3: Yes

Reviewer #4: (No Response)

Reviewer #5: Yes

Reviewer #6: Yes

5. Is the manuscript presented in an intelligible fashion and written in standard English?

Reviewer #2: Yes

Reviewer #3: Yes

Reviewer #4: (No Response)

Reviewer #5: Yes

Reviewer #6: Yes

6. Review Comments to the Author

Reviewer #2: Summary:

In this work, COVID-19 case data have been collected from around the world. The regions of waves are labelled. Also, XGBoost model is used to provide a minimum standard for future classifiers trained on this dataset. In addition, the utility of the dataset for the prediction of (future) waves.

The authors have addressed the raised comments.

Comments :

Abstract :

- - - - - - - - - - -

1 – The abstract is fine. No further comments .

Introduction Section :

- - - - - - - - - - - - - - - - - - - - - -

2 – This section is fine. No further comments .

Data and Methods Section :

- - - - - - - - - - - - - - - - - - - - - - - -

3 – This section is fine. No further comments .

Results Section :

- - - - - - - - - - - - - - - - - - - - - -

4 – This section is fine. No further comments .

Conclusion Section :

- - - - - - - - - - - - - - - - - - - - - -

5 – This section is fine. No further comments .

- - - - - - - - - - - - - - - - - - - - - - - - - - - - - - - - - - - - - - - - - - - - - - - - - - - - - - - - - - - - - - - - - - - - - - - - - - - - - - - - - - - - - - - - - - - - - - - - - - - - - - - - - - - - - - - - - - - - - - - - - - - - - - - - - - - - - - - - - - - - - - - - - - - - - - - - - - - - - - - - - - - - - - - - - - - - - - - - - - - - - - - - - - - - - - - - - - - - - - - - - - - - - - - - - - - - - - - - - - - - - - - - - - - - - - - - - - - - - - - - - - - - - - - - - - - - - - - - - - - - - - - - - - - - - - - - - - - - - - - - - - - - - - - - - - - - - - - - - - - - - - - - - - - - - - - - - - - - - - - - - - - - - - - - - - - - - - - - - - - - - - - - - - - - - - - -

Reviewer #3: This version of the manuscript is well improved. The authors have addressed all reviewer comments. The manuscript can be accepted for publication.

Reviewer #4: - The abstract is long and NOT satisfactory. It should contain the following parts:

i. The importance of or motivation for the research.

ii. The issue/argument of the research.

iii. The methodology.

iv. The result/findings.

v. The implications of the result/findings.

-where is keyword list.Authors should add keyword list contain of 5 to 8 keywords.

-The motivation and contribution need to be improve in introduction.Author should also add seprate paragaraph of orgnization at the end of introducation.

-Was the dataset balanced? if the dataset is unbalanced and may affect the results significantly. The authors

should solve the problem of the unbalanced dataset.

-Clearly highlight the mathematical terms used in the paper and explain them in the text.

-Author should discuss more recent reference in introducation

*COVID-19 detection by dogs: From physiology to field application-a review article

*Combination of Angiotensin (1-7) Agonists and Convalescent Plasma as a New Strategy to Overcome Angiotensin Converting Enzyme 2 (ACE2) Inhibition for the Treatment of COVID-19

*Derivatization and combination therapy of current COVID-19 therapeutic agents: a review of mechanistic pathways, adverse effects, and binding sites

*The next frontier in vaccine safety and VAERS: Lessons from COVID-19 and ten recommendations for action

- Conclusion to be made more systematic and future scope to be elaborated more on technical features

that are planned to be added in the proposed system in the near future.

- The use of English language is fine, however, it is recommended to be checked once again.

Reviewer #5: The authors have addressed all the comments. The manuscript is well structured. Abstract is okay. data collection, Methodology, results and discussion are okay. Relevant articles were cited and properly referenced.

Reviewer #6: (No Response)

7. PLOS authors have the option to publish the peer review history of their article (what does this mean?). If published, this will include your full peer review and any attached files.

Reviewer #2: No

Reviewer #3: No

Reviewer #4: No

Reviewer #5: **Yes: **Boluwaji Ade Akinnuwesi

Reviewer #6: No

---

## [Editor Report · Acceptance letter]

3 Apr 2023

PONE-D-22-27577R1 

COWAVE: A Labelled COVID-19 Wave Dataset for Building Predictive Models 

Dear Dr. Raman:

I'm pleased to inform you that your manuscript has been deemed suitable for publication in PLOS ONE. Congratulations! Your manuscript is now with our production department. 

Kind regards, 

on behalf of

Dr. Hilal Tayara 

Academic Editor

PLOS ONE